

# Mesospheric Anomalous Diffusion During Noctilucent Clouds

Fazlul I. Laskar[1], Gunter Stober[1], Jens Fiedler[1], Meers M. Oppenheim[2], Jorge L. Chau[1],
Duggirala Pallamraju[3], Nicholas M. Pedatella[4], Masaki Tsutsumi[5], and Toralf Renkwitz[1]

[1]Leibniz Institute of Atmospheric Physics, Kühlungsborn, Germany
[2]Boston University, Boston, USA
[3]Physical Research Laboratory, Ahmedabad, India
[4]National Center for Atmospheric Research, High Altitude Observatory, Boulder, Colorado, USA
[5]Arctic Environment Research Center, National Institute of Polar Research, Tokyo, Japan

**Correspondence:** Fazlul Laskar (laskar@iap-kborn.de)

**Abstract.** The Andenes specular meteor radar shows meteor-trail diffusion rates increasing on average by $\sim 20\%$ at times and locations where a lidar observes noctilucent clouds (NLCs). This high-latitude effect has been attributed to the presence of charged NLC but this study shows that such behaviors result predominantly from thermal tides. To make this claim, the current study evaluates data from three stations, at high-, mid-, and low-latitudes, for the years 2012 to 2016, comparing diffusion to

show that thermal tides correlate strongly with the presence of NLCs. This data also shows that the connection between meteor-trail diffusion and thermal tide occurs at all altitudes in the mesosphere, while the NLC influence exists only at high-latitudes and at around peak of NLC layer. This paper discusses a number of possible explanations for changes in the regions with NLCs and leans towards the hypothesis that relative abundance of background electron density plays the leading role. A more accurate model of the meteor trail diffusion around NLC particles would help researchers determine mesospheric temperature

and neutral density profiles from meteor radars.

# 1   Introduction

The motion and diffusion of meteor trails depends sensitively upon the properties of the neutral atmosphere where they ablate. Measuring meteor properties with radars enables researchers and weather modelers to estimate the state of the lower thermo-

sphere and upper mesosphere. Meteor radars most often observe underdense meteors in which the radar frequency exceeds the plasma frequency set by the peak meteor plasma density. Typically they have life-times that varies from 0.01-0.3 s at altitudes below 110 km. Studies of the meteor trail decay-time and effort to derive ambient temperatures from them have a long history (e.g., Greenhow and Neufeld, 1955; Murray, 1959), but even today there exist several subtle difficulties.

Theoretically the meteor trail diffusion (hereafter we refer it as MTD or $D_a$ should increase exponentially with altitude. However, the MTD derived from echo fading times measured by meteor radars deviate away from exponential behavior at





altitudes below about 85 km. Using chemistry based numerical simulation, Younger et al. (2014) reported that the deionization of the meteor trail by three-body attachment (a chemical process) at altitudes below 90 km could be responsible for the deviation. But, they were open to contributions from background dusts, such as meteor smoke particles and noctilucent cloud (NLC). Moreover, in a recent study Hocking et al. (2016) argued that the chemical processes are more important for the long

lived (non-underdense) meteors, where the importance of ozone chemistry has been discussed. A study by Singer et al. (2008) showed different behavior of the MTD coefficient profiles during NLC and non-NLC cases. They also noted that the strong and weak meteor based separation does show a partly similar behavior, so they could not conclude clearly the contributions from NLC. Here we investigate multiple years of NLC and MTD from different latitudes to investigate the lack of understanding in identifying the role of NLC and atmospheric dynamics.

Altitudinal profiles of temperature are essential for improved modeling of upper atmosphere dynamics at mesospheric heights. However, uninterrupted measurement of this parameter is not possible using traditional optical techniques due to cloud cover. If it were possible to derive temperature from MTD estimates, continuous temperature measurement could become a reality. Currently there are several difficulties in the deriving temperatures from meteor diffusion measurements as

there are several unknown and anomalous variabilities. Nevertheless, there are several techniques in use (e.g., Hocking et al., 1997; Hocking, 1999; Holdsworth et al., 2006; Stober et al., 2012; Holmen et al., 2016), which provide temperature estimates roughly at a cadence of about a day, but with their own merits and demerits.

## 2 Experimental Data

The primary data used for this investigation are from the specular meteor radars at Andenes (69°N,16°E) in Northern Norway,

Juliusruh (55°N,13°E) in Northern Germany, and Biak (1°S,136°E) in Indonesia. Other than these, NLC data from a Rayleigh-Mie-Raman (RMR) lidar in Andenes are also used to study the characteristics of meteor radar diffusion.

### 2.1 Specular Meteor Radar (SMR) Based Diffusion Coefficients

The most commonly observed meteors using a 32 MHz meteor radar are of the underdense type, for which the amplitude profile $A(t)$ decays approximately as per the following relation:

$$A(t) = A_0 \exp\{-(16\pi^2 D_a t)/\lambda^2\} = A_0 \exp\{-\ln 2\, t/\tau_{1/2}\} \tag{1}$$

where, $t$ is time, $D_a$ is ambipolar diffusion coefficient, $\lambda$ is wavelength of radar signal, and $\tau_{1/2}$ is the decay time to reach half of maximum amplitude ($A_0$):

$$\tau_{1/2} = \lambda^2 \ln 2/(16\pi^2 D_a) \tag{2}$$

Thus, knowing the decay rate $\tau_{1/2}$ from the meteor echo received, the ambipolar diffusion coefficient can be estimated. As

the number densities of the electrons in the meteor trail plasma are several orders of magnitude (at least 3 orders) greater than





the background plasma, the trail diffusion could be assumed as an approximation of the mesospheric neutral diffusion. This is because the movement of the trail positive ions are governed by neutrals through collisions.

We have estimated diffusion coefficient from such meteor decay rates for all the available years of meteor detections. But for the current study, based on the avilability of NLC data, only 4 years (2012-2016, excluding 2014) are investigated in details. Figure 1 shows the yearly composite (daily binned) $D_a$ values for all the available years of data obtained using the meteor radars located at low-, mid-, and high-latitudes stations. It can be seen here that, in general, the diffusion decreases with altitude until about 88 km, above which it starts increasing again. In the current study, meteors qualifying the following selection criteria are considered: (i) zenith angle less than 65 degrees, (ii) those during AE index less than 400 nT, and (iii) those having signal-to-noise-ratio (SNR) greater than 5 dB.

## 2.2 NLC data

The NLC data are obtained using the RMR lidar located at the Andoya (69°N,16°E) island in Northern Norway (Baumgarten, 2010), which is very close to the Andenes meteor radar site. Spectral and spatial filtering capability of this lidar enables continuous observations of NLC even during daylight conditions. Though the instrument existed for a long time, it had experienced several technical developments over the years. Since the year 2011 a pressure controlled single Fabry Perot etalon is used to filter out the background, which increased the SNR of the system (Fiedler et al., 2017). So, the NLC data used here are from the years 2012 to 2016 during clear sky hours of June-July-August over Andenes. The presence or absence of NLC are identified from integrated measurements, over about every 15 minutes intervals, during all the clear sky days.

## 3 Results

In the high-latitude summer mesosphere there occurs upwelling and the maximum of the upward motion lies close to the mesopause level (e.g., Smith, 2012; Laskar et al., 2017, and references therein). Due to such upward motion the summer mesosphere is the coldest region in the atmosphere. Figure 2 shows diffusion coefficients estimated from Whole Atmosphere Community Climate Model Data Assimilation Research Testbed for the year 2007 (WACCM+DART-2007) (Pedatella et al., 2014) temperature profiles over the stations. Assimilative temperatures are believed to provide close to realistic values as compared with satellite observations (e.g., Pedatella et al., 2014), but with better local time coverage. The conversion from temperature to diffusion is done using the simple relation $D = 6.39 \times 10^{-2} T^2 K_0/p$, where $p$, $T$, $D$, and $K_0$ are respectively pressure, temperature, diffusion, and zero field mobility factor. The value of the factor $K_0$ is debatable (e.g., Cervera and Reid, 2000; Hall et al., 2004) and we use $K_0=2.5 \times 10^{-4}$ m$^2$s$^{-1}$V$^{-1}$ (e.g., Meek et al., 2013; Younger et al., 2014). Here it may be noted that the diffusion derived from model follows the theoretically expected exponential law. But as mentioned above the observed diffusion from meteor radar based fading time shows deviation away from exponential behavior. Some investigations attributed such deviation to be due to deionization of the trail by three-body chemistry (Younger et al., 2014; Lee et al., 2013). But it may also be possible that the assumption of the ambipolar diffusion and Gaussian profile of meter trail radial plasma





distribution is too simple approximations, which needs further investigations.

From a comparison of Figures 1 and 2, one can say that the broad seasonal features showing altitude shift of constant value surfaces are similar, but the increased values at lower altitudes in summer differ in the datasets. This suggest that additional

physical processes are responsible for the MTD variability during summer.

It is well know that summer mesosphere at mid and high-latitudes are relatively colder locations in the atmosphere. Under such cold condition the saturated water vapor present and/or transported in the mesosphere freeze up and produce NLCs. NLCs are expected to remove free electrons and thus produce negatively charged ice particles. An earlier study by Singer et al. (2008)

used 6 days of meteor trail diffusion data and reported that the diffusion profiles have different behavior if separated based on the NLC presence or absence. In order to systematically investigate the role of NLC for larger datasets and for greater number of years, we have used Andenes RMR-lidar based NLC observation times to segregate the diffusion values. The leftmost column of Figure 3 shows such an NLC presence (yNLC) and absence (nNLC) based grouping for the measurements during clear sky days of June-July-August of the years 2012-2016, excluding the year 2014 wherein we had many data gaps for the

high-latitude station, Andenes. The horizontal histograms in the leftmost column represent the occurrences of NLC (total number of 15 minute intervals with NLC-presence) at a particular altitude for a particular summer. The middle and right columns in Figure 3 are for the MTD data from Juliusruh (mid-latitude) and Biak (low-latitude) SMRs, but they were segregated and then grouped based on the NLC sampling at Andenes. As the meteor trail diffusion at a particular altitude is log-normally distributed, the solid (for yNLC) and dashed (for nNLC) lines here are the geometric mean ($\bar{x} = exp[\overline{logX}]$) profiles and the

shaded regions represent their 99% confidence intervals (e.g., Ballinger et al., 2008). As there are reports that neutral density and thus MTD are influenced by geomagnetic activity (e.g., Yi et al., 2018), we have considered only those meteors that had occurred during relatively quiet geomagnetic conditions (AE index less than 400 nT).

From the grouping based on NLC occurrence, as shown in Figure 3, it can be seen clearly that there are differences between

diffusion profiles in the presence and absence of NLC. The high-latitude shows greater differences/separations than do the low-latitude. Physical causes of such anomalous behavior are discussed below.

## 4   Discussion

NLC particle sizes are of tens of nanometers and thus they are much heavier compared to ambient constituents. In the presence of such heavier particles, one may expect that a direct interaction with them, if any, would result in relatively smaller diffu-

sion compared to their absence. But what we see from the leftmost column of Figure 3 is the reverse, i.e., in the presence of NLC the SMR-radar measured diffusion coefficient gets enhanced. Here we present as list of possibility through which NLC may influence/modulate the meteor trail diffusion and in the following paragraphs we discuss in details about their role. (i) by capturing trail electrons thereby making the trail vanish faster in the eye of radar, (ii) by radiative heating due to presence



of semi-transparent NLC-layer, (iii) since NLC occurrence time shows a thermal tidal behavior, it may introduce a systematic artifact in our time sampling, (iv) neutral turbulences may sustain longer during the relatively colder NLC occurrence durations , which could help to diffuse the trail faster. (v) the NLC particles could absorb background free electrons thereby changing the electrodynamics of trail and background-plasma.

For (i), in the presence of NLC it may be expected that ice-particles absorb the trail electrons, which can lead to shorter life-time of the trail plasma. But the time constant of electron capture rate (order of seconds) (Rapp and Lübken, 2000) is longer than the typical life-time of the underdense trails (order of milliseconds). Also the abundance of NLC particles are at least 3 orders of magnitude less than trail electrons. Thus this process is very unlikely the cause for the enhanced diffusion.

For (ii), the radiative influence on the background atmosphere due to changes in the optical properties in the presence of NLC could increase the NLC particle temperature by 1 to $2°K$ (e.g., Espy and Jutt, 2002). As the number of NLC particles are very negligible compared to background neutral densities such rise in particle temperature would not contribute to the background temperature or diffusion.

To check if the anomaly during NLC could be occurring purely due to thermal tides, possibility (iii), we have used two additional stations; Juliusruh at mid-latitude (middle column in Figure 3) and Biak at low-latitude (rightmost column in Figure 3). But the local time sampling for the data grouping/classification has been taken as that from the high-latitude NLC occurrence. Even for the mid and low latitude data we can see that there exist difference between the two profiles in many of the years, e.g., in 2012, 2013, and 2016 for mid-latitude and in 2012, 2013, and 2015 for low-latitude. The presence of such differ-

ences/anomalies/enhancements for the majority of the cases in all the latitudes signifies that there is some systematic behavior in NLC occurrence, which is nothing but tidal (a local time dependent) behavior. From these multi-latitude dataset it is clear that the NLC based separation of diffusion coefficient also reflects the effect of thermal tide, which could arise because of the fact that the NLC occurrence show a tidal behavior (e.g., Fiedler et al., 2011; Gerding et al., 2013).

In order to investigate the tidal behavior in MTD, an hourly composite of the June-July, 2003-2017 diffusion coefficient data for the high-latitude station, Andenes is shown in Figure 4. Here it can be seen that the dominant variation is the diurnal tide, which is, in general, the strongest tide observed in the lidar dataset (e.g., Fiedler and Baumgarten, 2018), but presence of semidiurnal (two max./min.) can also be noted. This tidal behavior can also be seen in the histogram of the local time occurrence of NLC and no-NLC durations during June-July-August months, which is provided in Figure 5. A very clear diurnal

behavior in NLC occurrence can be seen in the year 2013, this could be the reason why the separations in Figure 3 are higher in this year for the Biak and Andenes station. Detailed discussion about tidal behavior in NLC can be found in Fiedler et al. (2011) and Fiedler and Baumgarten (2018). From Figure 3 it can be seen that the altitude of maximum separation between NLC and non-NLC diffusion profiles does not coincide with the altitude of maximum NLC occurrence, particularly in the years 2012 and 2013 where minor separation can be seen even at altitudes above NLC layer. This anomalous behavior could



also be attributed to additional contributions from tidal dependency of diffusion.

Another interesting fact from Figure 3 is that the yNLC and nNLC differences in MTD for the low-latitude location extend mostly at all the altitudes shown here, which is not the scenario for the high-latitude case, where these differences/enhancements

are of higher magnitude and are predominantly at lower altitudes, where the NLC does occur. From this different behavior of the low and high latitude MTD, we argue that there is tidal influence (as differences are seen in all latitudes), but in addition to that there are indications of significant contributions from NLC for high-latitude station. About the possibility (iv), Hall (2002) investigated the possibility of such mechanism to explain the deviations of diffusion away from the exponential behavior. However, on a later report (Hall et al., 2005) they ruled out such mechanism for radars having frequencies close to 30 MHz. They

also estimated that the turbulence diffusion in fact is lower in magnitude during summer than in winter. Using 10 rocket flights that were capable of high-resolution measurements of neutral density Lübken et al. (2002) argued that neutral turbulences are very weak during summer and the adiabatic lapse rate condition is hardly reached near the NLC layer. These earlier results imply that neutral turbulence is unlikely to be the cause for the enhanced diffusion during NLC.

For (v), in the absence of NLC the electrons in the trail could be short circuited by the background free electrons and thus this would reduce the effective ambipolar diffusion as the lost electrons would no longer contribute to the diffusion. But, when there is NLC they could absorb background electrons to reduce the density of the background free electrons, making a deficit to short-circuit the trail electrons. Under such condition the ambipolar diffusion of the meteor trail would be higher due to additional pressure from the electrons that are not short circuited as the background medium is less conductive. A schematic

cartoon for the background situation is depicted in Figure 6, where it can be seen that the background electrons are less in the NLC case (in right). This kind of explanation also suggest that the ambipolar diffusion assumption of the MTD is valid only when the background charges are very low compared to the trail electrons, similar to the situation as observed during the yNLC scenario. The possibility of such short-circuiting of the trail plasma by background free electrons was discussed both analytically and numerically by Dimant and Oppenheim (2006). This also suggest that for proper retrieval of the mesospheric

diffusion we would need an estimate of background electron density.

Changes in the background chemistry could also have an influence but at lower-altitudes where the reaction rates of the three body reactions are comparable to the life-time of meteor trail. This kind of explanation was used earlier to explain the reversal or turn around and then enhancement of MTD coefficient at lower altitudes (e.g., Lee et al., 2013; Younger et al., 2014). But, they did not rule out completely the importance of aerosols, such as NLC, meteor smoke.


For the high-latitude summer time data, Singer et al. (2008) had used the assumption of presence of neutral and charged dust, as was proposed by Havnes and Sigernes (2005), to explain the slower decay rate (i.e., higher diffusion as per Equation 2) in the presence of NLC. They also expected that the strong and weak meteors would be affected differently by the presence or absence of NLC. With their limited data from only 6 days, they showed that NLC and non-NLC diffusion behavior is, to

some extent, similar to diffusions during weak and strong meteor echoes. To investigate that if the enhancement during NLC




are affected by strong and weak meteor bias, we also have carried out a test in which all those meteors with SNR greater than 12 dB (strong meteors) were used and it was found that the NLC and non-NLC difference scenario still persist as in Figure 3, though they get narrower as the error limit increases due to lesser number of meteors. The test case figure is provided in the supplementary information Figure S1. This test also implies that the diffusion from weaker meteor could be more anomalous

and it adds credence to our hypothesis presented in the previous paragraph.

From this anomalous behavior of the meteor radar diffusion during NLC occurrence it is clear that some of the temperature profile estimation methods which uses standard pressure levels will yield misleading results at lower altitudes in presence of NLC. It also indicates that the use of MTD reversal altitude as constant density surface would not be valid under NLC

conditions, unless the NLC contribution has been deciphered. Further, for the derivation of temperature at NLC altitudes from SMR-diffusion measurements, proper retrieval algorithm considering the NLC related anomaly is very important. Such retrieval would need information about background electron density, the size of NLC particles, their charge state (Chau et al., 2014) and is a subject of future studies.

## 5    Conclusions

Meteor trail diffusion variations measured by SMRs at high- (Andenes), mid- (Juliusruh), and low- (Biak) latitude stations, have been used to investigate the mesospheric diffusion variability during summer season. The Andenes SMR based diffusion coefficient during NLC has been found to be enhanced compared to no-NLC durations. Applying the NLC occurrence based local time sampling as that of the high-latitude to the mid- and low-latitude SMR based diffusions some enhancements are seen but are of lower magnitudes, indicating general tidal influence. This is because the NLC occurrence has a tidal modulation

and thus the meteor samplings are biased by it. The tidal behavior in both NLC occurrence and SMR based diffusion have been found to be dominated by diurnal tide. But in addition to the tidal influence, which influences all altitudes in this limited region, for the high-latitude station we see that the enhancements are of higher magnitude and predominantly at NLC occurring altitudes. This suggests that in addition to tides NLC also influences the SMR diffusion.

The NLC particles could absorb many of the background free electrons to create lesser conducting background medium. Based on current results it is hypothesized that under such background electron deficit situation the trail diffusion would be enhanced as there are lesser number of free electrons in the background to short-circuit the trail electrons. But in the absence of NLC the relatively higher number of background free electrons would help to short circuit electrons from trail thereby reducing the ambipolar diffusion. From this statistical study of the anomalous behavior of SMR based diffusion measurements

we conclude that the temperature estimations from them would need a detailed retrieval algorithm to account for the influence of background electrons, ice particles and other dusts/aerosols.



*Data availability.* The meteor mpd data of the Andenes and Juliusruh systems can be made available online at ftp://ftp.iap-kborn.de upon request. The Biak system data are available at IUGONET (http://www.iugonet.org). The RMR-lidar based NLC data can be obtained from JF. WACCM+DART-2007 data can be obtained from NP. The National Center for Atmospheric Research is sponsored by the U.S. National Science Foundation.

5   *Author contributions.* FIL and GS conceived the preliminary idea. FIL analyzed most of the presented data in coordination with GS, JLC, and JF. JLC also contributed by comparing the results with data from another independent analysis. MMO helped in the interpretation of the results and provided useful comments on presentation of the results. NMP provided the WACCM+DART data. DP, MT, and TR participated in discussions related to interpretation and presentation of the results. All authors read and approved the final version of the manuscript.

*Acknowledgements.* We acknowledge the support of the IAP staff for keeping the radars running. This work is partially supported by the
10   WaTiLa project (SAW-2015-IAP-1 383). Data acquisition of meteor radar at Biak has been done by Research Institute for Sustainable Humanosphere (RISH), Kyoto University. Distribution of the data has been partly supported by the IUGONET (Inter-university Upper atmosphere Global Observation NETwork) project (http://www.iugonet.org/) funded by the Ministry of Education, Culture, Sports, Science and Technology (MEXT), Japan. The National Center for Atmospheric Research is sponsored by the U.S. National Science Foundation. We thank prof. Cesar La-Hoz of Arctic University of Norway for his discussion and useful comments.





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





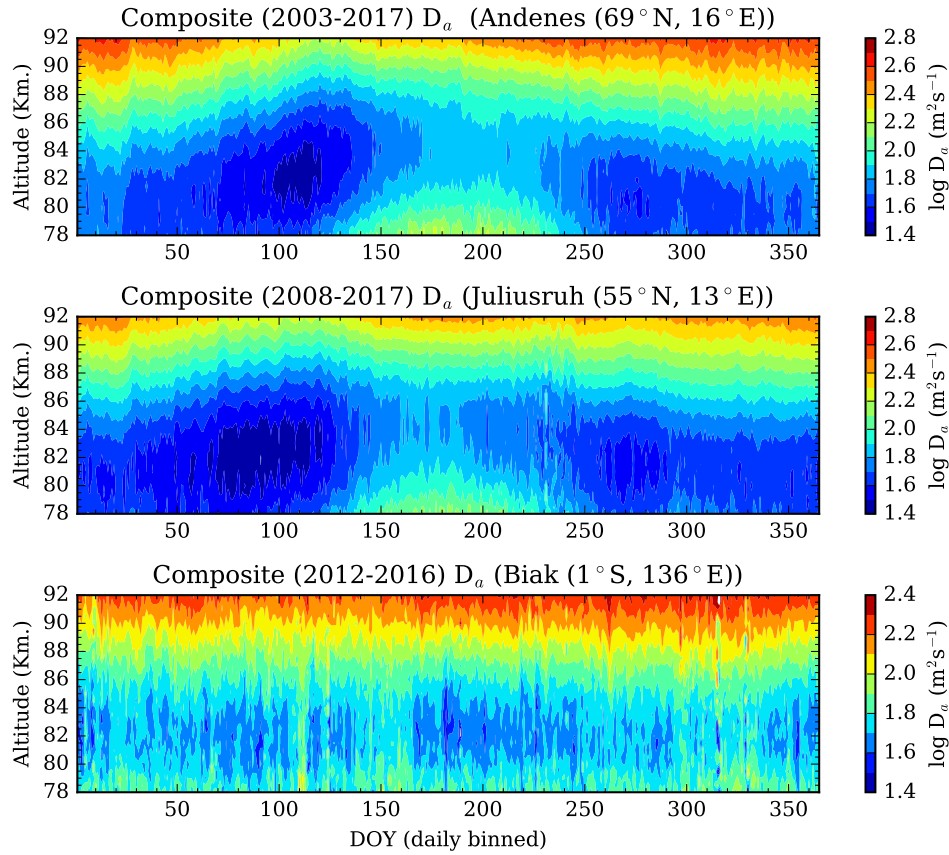

**Figure 1.** Diffusion coefficient ($D_a$) measured by SMRs located at high- (Andenes, 69°N, upper), mid- (Juliusruh, 55°N, middle), and low-latitude (Biak, 1°S, bottom) are shown. Notable features like increased $D_a$ at lower altitudes are protruding out in the mid- and high-latitude stations during summer.



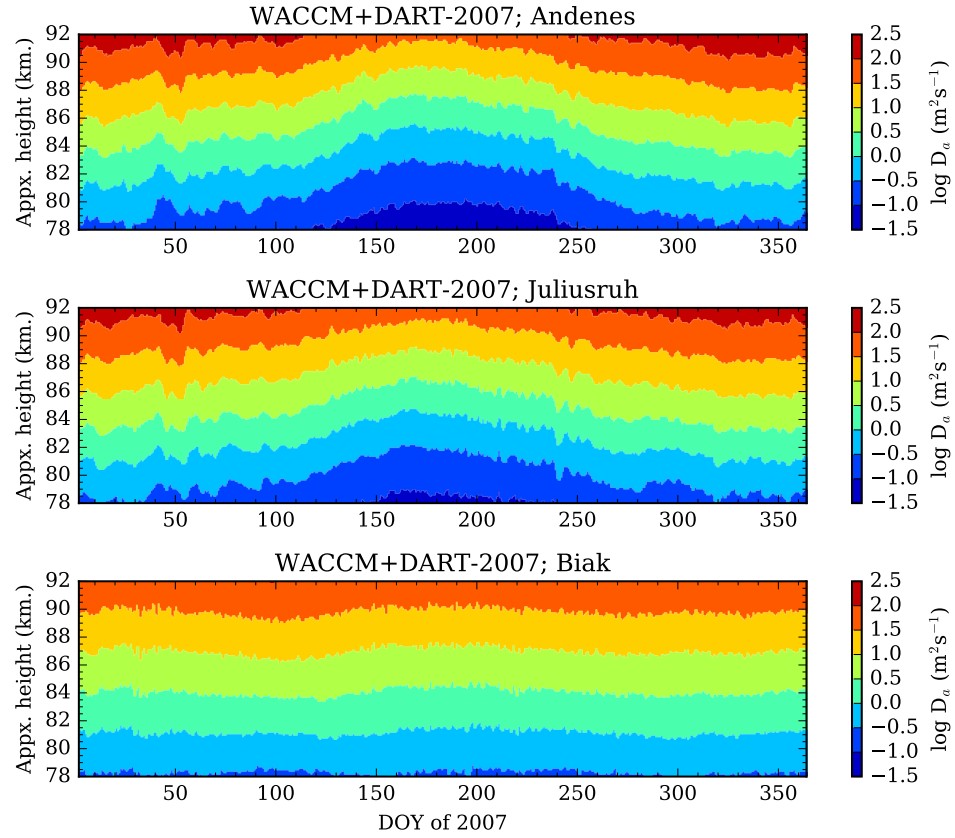

**Figure 2.** Representative yearly diffusion values obtained by directly converting the WACCM+DART 2007 temperatures over the 3 stations.
They show nearly similar seasonal variability, except the increased meteor trail diffusion at lower altitudes seen in Figure 1. Also the summer
enhancement seen in 1 is not visible. This implies that the enhanced diffusion has to do with things other than temperature variability.



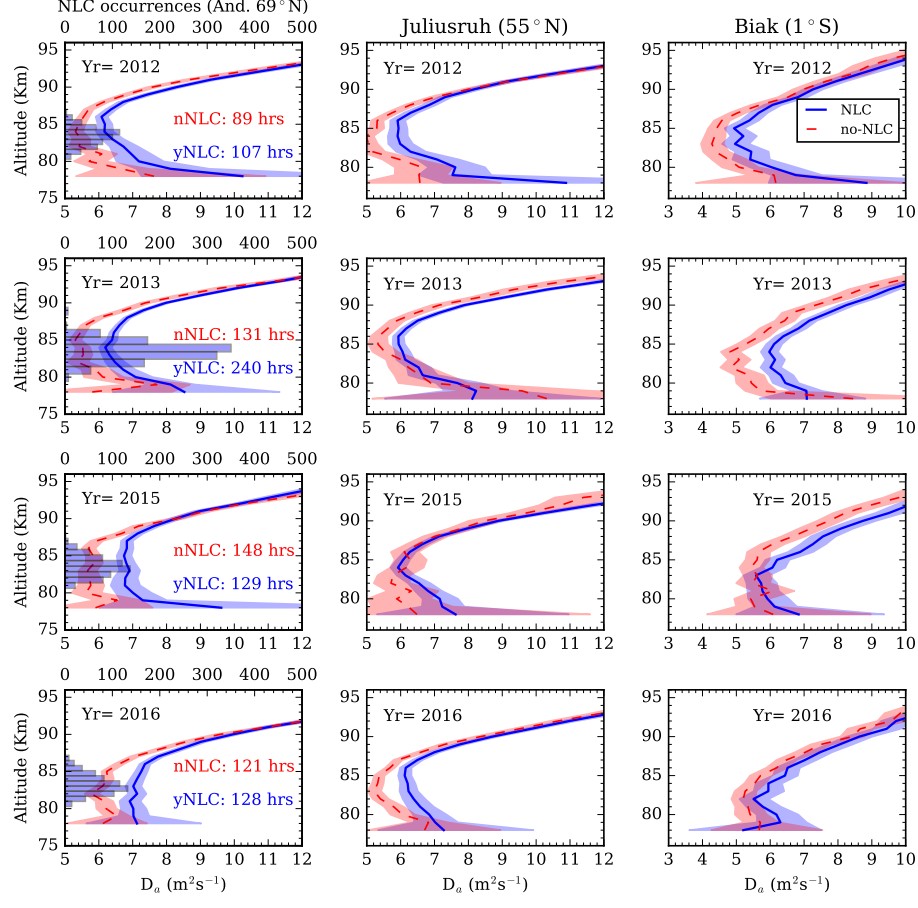

**Figure 3.** Mean meteor trail diffusion coefficients after segregating them based on presence of NLC (yNLC, blue) and no-NLC (nNLC, red) over Andenes station (leftmost column) are shown. Using the time sampling from NLC occurrences at Andenes, the $D_a$ measurements at mid-latitude, Juliusruh (middle column) and low-latitude, Biak (right column) are also grouped. The shaded regions around the averaged vertical profiles, dashed for nNLC and continuous for yNLC, are the 99% confidence intervals. The histograms in the top axes of leftmost column show the altitude variability of NLC occurrences measured using RMR-lidar at Andenes during June-July-August months. Notable features are: (i) $D_a$ during yNLC is enhanced compared to nNLC, (ii) NLC based grouping also show separations/enhancements at mid- and low-latitudes, and (iii) the high-latitude enhancements are higher and are predominantly at lower altitudes, while the low-latitude ones extend over all the altitudes presented here.





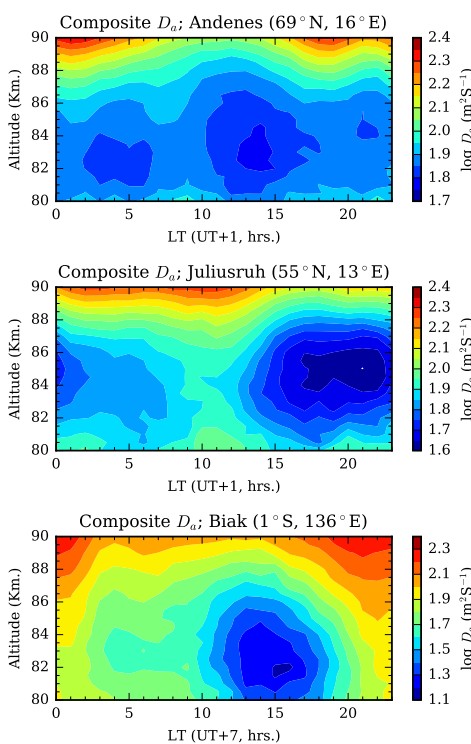

**Figure 4.** Composite $D_a$ during June-July of the years 2012-2016, excluding 2014 over the three stations. It can be seen that the diurnal tide (one maximum/minimum) is the most dominant component, while the presence of semidiurnal (two maxima/minima) tide could also be recognized for the Andenes. This signifies that there is a dominant tidal dependence in both NLC occurrence and diffusion variability.




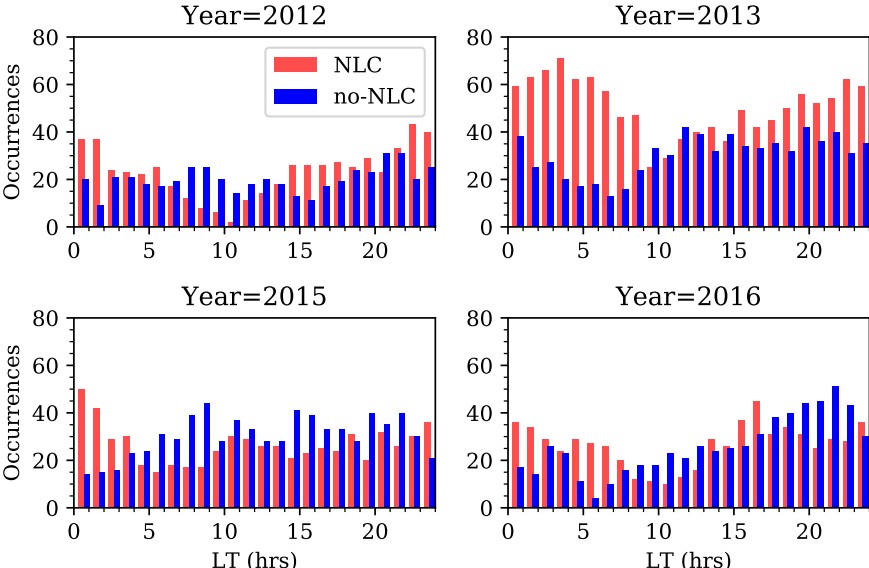

**Figure 5.** NLC and no-NLC occurrences (number of 15 mins. intervals) over local time during the observation years are shown. These samplings show that the NLC and no-NLC occurrence/sampling are not uniform over local-time hours. Further detailed discussion and analysis about the tidal dependence of NLC occurrence could be found in Fiedler et al. (2011).

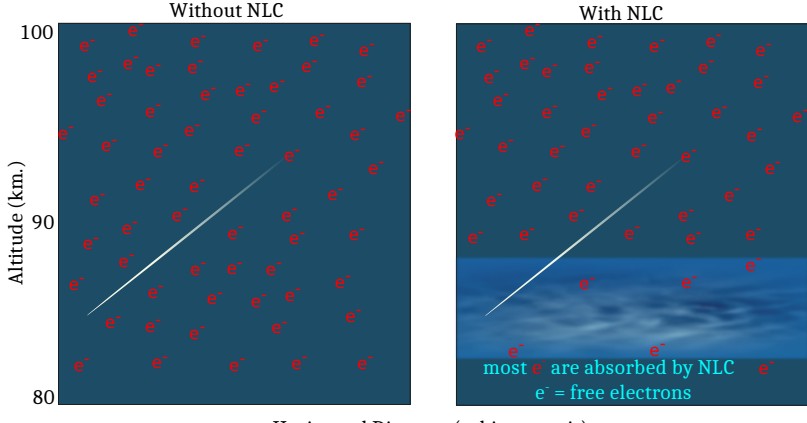

**Figure 6.** A schematic illustration for the background state without-NLC (in left) and with-NLC (in right) is shown. In the with-NLC case background electrons at lower altitudes are mostly taken up by the NLC particles creating a deficiency of electrons, which therefore cannot take part in short-circuiting the trail electrons and thus the meteor radar measured diffusion appears to be enhanced.