# Peer review of "Mesospheric Anomalous Diffusion During Noctilucent Clouds"

_Atmospheric Chemistry and Physics, 2018_

## Referee Comment (RC1) · Anonymous Referee #1 · 27 Dec 2018

The authors reported the difference for Da measured by the meteor radars during the existence of NLCs and considered the possible mechanism related with the observations. However, the deduced conclusions from the analysis seemed to be more clarified before publication. My main concerns are listed as follows:

1. The paper used daily Da, which is proportion to the T and P, and can be obtained from satellite observations (such as SABER or MLS). Using the Da from satellite measurements during the same period, i.e., 2012-2016 should be better than WACCM-DART data during 2007.

2. Figure 3, the authors claim the obvious difference of Da during yNLC/nNLC for high-,middle- and low- latitude stations. Is the result statistically significant? If using a random sampling during the lidar observation period to re-group the yNLC and nNLC,

how about the response of Da at different latitudes?

3. The Da in Figure 3 is separated according to the lidar measurements. The lidar has time resolution of 15 min. Are the Da measured by radar at different location fully covered the lidar sampling period? For example, are the Da at Andenes, Juliusruch and Biak all available for 107/89 hrs of yNLC/nNLC period during 2012 (the first row of Figure 3)? If not, what's the proportion of the data coverage?

4. The authors indicate the global tide are responsible for the observed difference at different latitudes. However, (1) the dominant tidal model also depends on the latitudes. For example, the semi-diurnal tide is dominant at high latitude, while, diurnal tide is dominant at low latitude. This situation can also be found in Figure 4. (2) is the local time response of NLC and tide correlated? In Figure 5, except for the year of 2013, the NLC did not show significant local time dependence. According to the comments above, I am a little confused, since the NLC did not show significant local time dependence, how to explain the observed difference at middle and low latitude region? Especially under the scenario of the different dominant tidal modes and different local time dependence for different latitude?

---

## Referee Comment (RC2) · Anonymous Referee #2 · 4 Feb 2019

The manuscript is dedicated to study of the relation between NLC events and ambipolar diffusion behavior at heights of mesopause. The subject is quite interesting as well as results, but there are some questions before publication.

1. The authors report the difference between mean $\log_{10}(D_a)$ profiles for yNLC and nNLC. They used three stations (two in region of NLC - Andenes (69N,16E), Juliusruh (55N,13E)) and one is out of that region - (1S,136E)). So one can expect significant difference between profiles for yNLC and nNLC for midlatitude stations and no difference for equatorial. Accords to fig. 3 we can see differences for all three stations.

2. NLC maximum is at height near 83km (fig.3). But significant distinction in Da for NLC and non-NLC time can be seen at lower heights. Why? Juliusruh Da profiles show less affection of NLC effect than Andenes. Why? Juliusruh is situated in middle of band of

[Figure]

NLC occurance so we have to expect major effect?

3. The manuscript is dedicated to revealing the connection between NLC and Da. However, the major affection (as authors admitted) to Da for proposed segregation is due to temperature oscillations. It's good idea to exclude diurnal and semidiurnal oscillation (for ex. with help of harmonic fitting) from produced times series of log10(Da). After that one can expect removing the affection of "temperature" and we will see pure results.

4. The authors consider the total relation between ambipolar diffusion coefficient and half decay time of meteors by skiymet radar (eq. 2). But it maybe not totally correct. Some effects may bias estimates of log10(Da) to greater values. Besides, height determination at edge of meteor band near 80km and 100km is quite unreliable due to possible jumps from middle of meteor band (90km) due to ambiguity of phase measurements. Thus significant increase of log10(Da) at lower heights seems to unreliable. Besides, the RMS error is increasing here. Why? Looks like distribution of log10(Da) at these heights is quite wide and not Gaussian. What's the result of simple averaging of log10(Da) in this case?

5. Experimental Data. No references to descriptions, no explanation. Just "data". Why should one know what those data are? How did they get them? How processed? Ok, Andenes and Juliusruh MR are quite familiar and results are already published before. As for Biak, results are quite rare as well as data maybe quite unreliable. I have downloaded MPD data from http://database.rish.kyoto-u.ac.jp/arch/iugonet/mwr_bik/index_mwr_bik.html and found a great percentage of ambiguous meteors. It says about illness of system. Such data should be used carefully.

6. NLC mainly observed in mid-latitudes (43-65 latitude's degree, or 50-70 latitude's degree by other sources). Why to use lidar for detection of NLC located in high-latitude (69N)? How it affects on detection of NLC? How it affects on segregation for other stations (Juliusruh and Biak)? In other words, if we see NLC event at current time at

certain station should we expect it at other stations in same hours?

---

## Author Comment (AC1) · 12 Mar 2019

We thank the reviewer for the thoughtful comments and suggestions. Below we answer them individually. The authors' response start with "Response".

The authors reported the difference for Da measured by the meteor radars during the existence of NLCs and considered the possible mechanism related with the observations. However, the deduced conclusions from the analysis seemed to be more clarified before publication. My main concerns are listed as follows:

1. The paper used daily Da, which is proportion to the T and P, and can be obtained from satellite observations (such as SABER or MLS). Using the Da from satellite measurements during the same period, i.e., 2012-2016 should be better than WACCM-

[Figure]

DART data during 2007.

Response: We thank the reviewer for pointing at the satellite observations. However, because of the following issues we refrain ourselves in their use for the current study:

(i) The reviewer might be aware that MLS being an A-train (afternoon at equator) satellite has just one fixed local time over a location, which also does not change much over couple of days for SABER's case. Thus they are not necessarily coincident with the lidar NLC observations. Also because of their low (for SABER) and no (for MLS) local time shift they will be highly modulated by tides.

(ii) Further one has to consider how both satellites retrieve the parameters of T and p. In the case of MLS the observed irradiances are used to derive temperature and geopotential height. The state vector uses 47 fixed pressure levels and a geopoetential reference height at 100 hPa (Schwartz et al., 2008). Due to the coarse vertical resolution of MLS at the MLT region a precise observation of T and p is not achievable considering the required accuracy.

In the case of SABER the retrieval of T and rho or p are not independent. The primary observed quantity is irradiance where density of a certain Molecule, $CO_2$ in the mesosphere is converted into a neutral density assuming a volume mixing ratio. Considering the statistical errors of this conversion (Remsberg et al., 2008, Rezac et al., 2015), unfortunately does not hold the required accuracy. The second issue is that we want to study a polar effect, which is even more challenging with SABER due to the Yaw cycle, which would further deplete our measurement statistics.

However, as pointed out by the reviewer it is might be worth to collect more data and compare just the times of the satellite overpasses. Considering the present statistical database it is not sufficient to do that. Such comparisons seem to be more beneficial for systems at lower latitudes, with a much better instrumental coverage, in particular, from the SABER instrument.

References: Schwartz, M. J., et al. (2008), Validation of the Aura Microwave Limb Sounder temperature and geopotential height measurements, J. Geophys. Res., 113, D15S11, doi:10.1029/2007JD008783.

Remsberg, E. E., et al. (2008), Assessment of the quality of the Version 1.07 temperature-versus-pressure profiles of the middle atmosphere from TIMED/SABER, J. Geophys. Res., 113, D17101, doi:10.1029/2008JD010013.

Rezac, L., Y. Jian, J. Yue, J. M. Russell III, A. Kutepov, R. Garcia, K. Walker, and P. Bernath (2015), Validation of the global distribution of $CO_2$ volume mixing ratio in the mesosphere and lower thermosphere from SABER, J. Geophys. Res. Atmos., 120, 12,067–12,081, doi:10.1002/2015JD023955.

2. Figure 3, the authors claim the obvious difference of Da during yNLC/nNLC for high-,middle- and low-latitude stations. Is the result statistically significant? If using a random sampling during the lidar observation period to re-group the yNLC and nNLC, how about the response of Da at different latitudes?

Response: We thank the reviewer for this suggestion. While carrying out this test we have come across a bug in our program, which removed many of the meteor trails having extreme values of diffusion from only NLC case and not from no-NLC case of analysis. After making this correction, we see that the difference between NLC and no-NLC profiles are only significant at high-latitudes. At mid- and low-latitudes they are either not-systematic or within the 95% significance levels. As per suggestions of the reviewer, we did a random sampling during the lidar observation period and also during whole summer (June-July-August). In both cases there were no such difference between the profiles at over any of the stations/latitudes. Results from the first test using lidar observation interval are shown in illustration figure 1 (attached) or in supporting info Figure S2 of the revised manuscript. Some discussions on this are added in P.6 L.17-22 in the revised MS.

3. The Da in Figure 3 is separated according to the lidar measurements. The lidar

has time resolution of 15 min. Are the Da measured by radar at different location fully covered the lidar sampling period? For example, are the Da at Andenes, Juliusruch and Biak all available for 107/89 hrs of yNLC/nNLC period during 2012 (the first row of Figure 3)? If not, what's the proportion of the data coverage?

Response: Since lidar observations needs clear sky (no cloud) condition and radar do not have such restrictions the Da measurements durations available are much higher than lidar. However, as there are some datagaps in the radar observations and also we have not considered any intervals where the geo-magnetic activity was high (AE>400 nT) the Da measured at different location do not fully cover the lidar observation windows. Now, based on the common availability of NLC and Da data we have revised Figure 3 and Figure 5. Now in the revised Figure 3 all the common available durations are depicted, where one can see that the common windows are different at different latitudes. Also, in the revised Figure 5 we have not considered the NLC observations where we have no Da measurements. About the proportion, the numbers are now depicted in Figure 3. Some discussions on Da availabiltiy over different stations are mentioned in P6 L.8-15.

4. The authors indicate the global tide are responsible for the observed difference at different latitudes. However, (1) the dominant tidal model also depends on the latitudes. For example, the semi-diurnal tide is dominant at high latitude, while, diurnal tide is dominant at low latitude. This situation can also be found in Figure 4. (2) is the local time response of NLC and tide correlated? In Figure 5, except for the year of 2013, the NLC did not show significant local time dependence. According to the comments above, I am a little confused, since the NLC did not show significant local time dependence, how to explain the observed difference at middle and low latitude region?Especially under the scenario of the different dominant tidal modes and different local time dependence for different latitude?

Response: We thank the reviewer for this thoughtful suggestion in (1). We have now added couple of sentence on this in P.5 L.30-32. On (2), yes, we agree that even in

the revised Figure 5 NLC does not have very strong diurnal variations, except in the year 2013. As mentioned above there was a bug in the processing program, which gave rise to systematic difference between Da profiles during NLC and no-NLC at all latitudes. Now with the correction such differences exist only at high-latitudes. Thus in the revised manuscript we re-interpret out corrected results and state that: As the NLC data used here does not show significant daily variations we interpret the observed differences at high-latitudes are primarily due to NLCs. We hope that the reviewer finds our revised results more convincing.

---

## Author Comment (AC2) · 12 Mar 2019

We thank the reviewer for the thoughtful comments and suggestions. Below we answer them individually. The authors' response start with "Response".

The manuscript is dedicated to study of the relation between NLC events and ambipolar diffusion behavior at heights of mesopause. The subject is quite interesting as well as results, but there are some questions before publication.

1. The authors report the difference between mean log10(Da) profiles for yNLC and nNLC. They used three stations (two in region of NLC - Andenes (69N,16E), Juliusruh (55N,13E)) and one is out of that region - (1S,136E)). So one can expect significant difference between profiles for yNLC and nNLC for midlatitude stations and no difference

for equatorial. Accords to fig. 3 we can see differences for all three stations.

Response: Yes, from a separation based on yNLC and nNLC that is what expected. Before answering this we would like to excuse that during data analysis of the previous manuscript we made a mistake. The mistake was that, in the previous analysis we had removed some meteors having extreme diffusion values using a 3 sigma filtering. This however was done only for the nNLC case and by mistake was not performed for the yNLC case. This has led to such systematic shift between yNLC and nNLC profiles at all the latitudes.

After correction of the above mistake/bug, we see that the yNLC and nNLC based separation exist only at high-latitudes and not at mid- and low-latitudes (please see revised Fig. 3). At mid-latitudes (Juliusruh), the NLC occurs on rare occasions (Nielsen et al., 2011). A quantitative estimation by Hervig et al., (2016) showed that the NLC at mid-latitude are at least 5-times weaker than those at high-latitudes. So, from the NLC based separation point of view, the mid-latitude is roughly similar to low-latitudes. Thus, as per expectation such separation is observed only at high-latitude, which can be seen in Figure 3.

References: Nielsen, K., G. E. Nedoluha, A. Chandran, L. C. Chang, J. Barker-Tvedtnes, M. J. Taylor, N. J. Mitchell, A. Lambert, M. J. Schwartz, and J. M. Russell (2011), On the origin of mid-latitude mesospheric clouds: The July 2009 cloud outbreak, J. Atmos. Sol. Terr. Phys., 73(14-15), 2118–2124.

Hervig, M. E., Gerding, M., Stevens, M. H., Stockwell, R., Bailey, S. M., Russell III, J. M., and Stober, G.: Mid-latitude mesospheric clouds and their environment from SOFIE, J. Atmos. Sol.-Terr. Phy., 149, 1–14, https://doi.org/10.1016/j.jastp.2016.09.004, 2016.

2. NLC maximum is at height near 83km (fig.3). But significant distinction in Da for NLC and non-NLC time can be seen at lower heights. Why? Juliusruh Da profiles show less affection of NLC effect than Andenes. Why? Juliusruh is situated in middle of band of

NLC occurance so we have to expect major effect?

Response: In the revised manuscript (Figure 3) such differences are seen predominantly at NLC maximum altitudes. Some significant differences are also seen over the NLC peak, which may have some contributions from thermal tides. As NLC occurs very rarely over Juliusruh, such differences are below 95% significance level as can be seen in the revised Figure 3.

3. The manuscript is dedicated to revealing the connection between NLC and Da. However, the major affection (as authors admitted) to Da for proposed segregation is due to temperature oscillations. It's good idea to exclude diurnal and semidiurnal oscillation (for ex. with help of harmonic fitting) from produced times series of log10(Da). After that one can expect removing the affection of "temperature" and we will see pure results.

Response: As we mentioned above, in the revised analysis, after correcting the bug in the analysis program, we see that the tidal effects are very weak. Moreover, during a colder phase of thermal tide diffusion is expected to be lower, however, what we observe is the reverse, in other words, cold phase is expected to increase NLC production, which should lead to enhanced diffusion. So, we conclude that the effect that we observe during NLC is not from thermal effects but from electrodynamic interactions between trail and background electrons.

Also, the tidal affects are taken into account as we compare the occurrence rates of NLC times and the observations without NLC. There is a weak daily pattern visible in Figure 5 that could be removed using the suggested harmonic fit. Further, we inspected individual days and due to the day-to-day tidal variability a harmonic fit is not going to lead to a much better removal of tidal effects. However, the reviewer is right that a potential tidal effect could reduce or even increase the reported effect. Looking at a model (LIMA, NAVGEM-HA) results of tidal amplitudes typically 1-3 K are expected for the semi-diurnal tide and up to 5 K at 90 km altitude for the diurnal tide. However, as

mentioned before, there is a huge local time variability of the tides and tidal amplitudes that are difficult to be removed.

4. The authors consider the total relation between ambipolar diffusion coefficient and half decay time of meteors by skiymet radar (eq. 2). But it maybe not totally correct. Some effects may bias estimates of log10(Da) to greater values. Besides, height determination at edge of meteor band near 80km and 100km is quite unreliable due to possible jumps from middle of meteor band (90km) due to ambiguity of phase measurements. Thus significant increase of log10(Da) at lower heights seems to unreliable.

Response: The reviewer brings up two questions about the analysis procedure. The half decay time from the meteor radar is obtained according to Hocking et al., [2001]. Without going into the details of signal processing the reviewer is right that the absolute value might not be true. However, the applied method is consistently used in all the analysis and whatever offset exists in the absolute value does not map through to the relative comparison presented herein. The authors do not claim to infer absolute values of ambipolar diffusion coefficients based on the half decay time. Recent solutions, invoking a full wave scattering model of an ambipolar diffusing plasma, indicate that the derivation of absolute diffusion coefficients may remain illusive and is not solved by signal processing alone for most of the present systems.

The range estimation mentioned by the reviewer is no longer of relevance herein. The Andenes and Juliusruh meteor radar are operated at a pulse repetition frequency of 625 Hz leading to an unambiguous range determination of 220 km (two way monostatic backscatter case) or in other words only low elevation meteors might fold into the meteor layer as referred to by the reviewer. We did not include meteors below 65° off zenith angle in our analysis to avoid this type of contamination. Further, we conduct a cleaning of folded meteor echoes due to interferometric issues at low elevation angles. These meteors are usually folded to near zenith angles and have large phase errors. These meteors are removed as well. Some discussion on these lines are added in P2 L 25-30.

Only in the year 2012 and 2013 the radars used a higher pulse repetition frequency. However, as the profiles shown in Figure 3 indicate, there is no jump between the mean profiles. Similar features are also reported in Younger et al., 2015 (GRL) with an ATRAD type meteor radar, which are usually operated at even lower pulse repetition frequencies.

Reference: Younger, J. P., I. M. Reid, R. A. Vincent, and D. J. Murphy (2015), A method for estimating the height of a mesospheric density level using meteor radar, Geophys. Res. Lett., 42, 6106–6111, doi:10.1002/2015GL065066.

Besides, the RMS error is increasing here. Why? Looks like distribution of log10(Da) at these heights is quite wide and not Gaussian. What's the result of simple averaging of log10(Da) in this case?

Response: The error at the edges (below 80 km or above 95 km) are increasing due to the lesser number of meteors. A version of the Figure 3 with simple averaging of log10(Da) is shown below (Illustration 1). As the behavior of the plots and differences are similar to that in Figure 3, we keep the Da plots in Figure 3 in the manuscript.

5. Experimental Data. No references to descriptions, no explanation. Just "data". Why should one know what those data are? How did they get them? How processed? Ok, Andenes and Juliusruh MR are quite familiar and results are already published before. As for Biak, results are quite rare as well as data maybe quite unreliable. I have downloaded MPD data from http://database.rish.kyoto-u.ac.jp/arch/iugonet/mwr_bik/index_mwr_bik.html and found a great percentage of ambiguous meteors. It says about illness of system. Such data should be used carefully.

Response: The reviewer made a good suggestion. We added a few lines on how the radars are operated and referring to the analysis presented in Hocking et al., 2001. However, a detailed description of the data signal processing might not be helpful and does not provide a significant improvement to the results. The Andenes and Juliusruh MR well documented radars. However, as suggested by the reviewer, we add a few

lines pointing out why Biak is more complicated and the data has to be used with more care. A reference (Batubara et al., 2018) containing technical and other details of the Biak system is also added in the revised manuscript. Possible range aliasing issue in Biak system are added in P2 L 29-32.

6. NLC mainly observed in mid-latitudes (43-65 latitude's degree, or 50-70 latitude's degree by other sources). Why to use lidar for detection of NLC located in high-latitude (69N)? How it affects on detection of NLC? How it affects on segregation for other stations (Juliusruh and Biak)? In other words, if we see NLC event at current time at certain station should we expect it at other stations in same hours?

Response: In the past there were some comparison of NLC occurrence rates with local measurements at mid-latitudes and satellites (SOFIE, Hervig et al., 2016). The NLC occurrence rate drops off by a factor of 5 between the polar latitudes around Andenes and the mid-latitudes at Juliusruh. Hence, the effect should be most pronounced at high-latitudes. Advection of NLC over large horizontal distances can indeed link higher latitudes with the mid-latitudes (Kaifler et al., 2018). However, the main reason to compare Juliusruh and Andenes are to delineate potential tidal affects. Cimatologies of tides indicate (at least for the wind) that tidal pattern is rather similar between Andenes and Juliusruh (Pokhotelov et al., 2018). This is also the reason why the Biak station was included. However, there is no physical reason why an NLC should be seen at all three stations at the same time.

References: Pokhotelov, D., Becker, E., Stober, G., and Chau, J. L.: Seasonal variability of atmospheric tides in the mesosphere and lower thermosphere: meteor radar data and simulations, Ann. Geophys., 36, 825-830, https://doi.org/10.5194/angeo-36-825-2018, 2018.

Kaifler, N., Kaifler, B., Wilms, H., Rapp, M., Stober, G., & Jacobi, C. (2018). Mesospheric temperature during the extreme midlatitude noctilucent cloud event on 18/19 July 2016. Journal of Geophysical Research: Atmospheres, 123, 13,775–13,789.

https://doi.org/10.1029/2018JD029717

Please also note the supplement to this comment:
https://www.atmos-chem-phys-discuss.net/acp-2018-1028/acp-2018-1028-AC2-supplement.pdf

[Figure]

Illustration 1: A version of Figure 3 with simple averaging of the log10(Da).

**Fig. 1.**

**Supplement:**

[revised manuscript text omitted]

---

## Author Response (AR2)

List of changes/corrections:

1) A spelling mistake was corrected in Figure 6: Details: "Arbitraty" changed to "Arbitrary"
2)  "...are are…." at last line of in Figure 3 caption has been changed to "…..are…."
3) An additional affiliation for the author Masaki Tsutsumi has been added.